# Unsupervised Out-of-Distribution Detection with Batch Normalization

## Abstract

Likelihood from a generative model is a natural statistic for detecting out-of-distribution (OoD) samples. However, generative models have been shown to assign higher likelihood to OoD samples compared to ones from the training distribution, preventing simple threshold-based detection rules. We demonstrate that OoD detection fails even when using more sophisticated statistics based on the likelihoods of individual samples. To address these issues, we propose a new method that leverages batch normalization. We argue that batch normalization for generative models challenges the traditional *i.i.d.* data assumption and changes the corresponding maximum likelihood objective. Based on this insight, we propose to exploit in-batch dependencies for OoD detection. Empirical results suggest that this leads to more robust detection for high-dimensional images.

## 1 Introduction

Modern neural network models are known to make poorly calibrated predictions (Guo et al., 2017; Kuleshov et al., 2018), and can be highly confident even for unrecognizable or irrelevant inputs (Nguyen et al., 2015; Moosavi-Dezfooli et al., 2017). This has serious implications for AI safety (Amodei & Clark, 2016) in real world deployments, where a model could receive inputs that are beyond its training distribution. Detecting examples that are out of the training distribution becomes a viable solution: when encountering such samples, the model could choose to provide low confidence estimates or even abstain from making predictions (Cortes et al., 2017).

Density estimation is one approach to detecting out-of-distribution (OoD) samples. A likelihood-based model is trained on the input samples; during evaluation, samples that have low likelihoods are assumed to be out-of-distribution. For high-dimensional inputs (such as images), deep generative models have been able to generate realistic samples as well as achieving good compression capabilities, which indicates high likelihoods on the training distribution (Ballé et al., 2016; Kingma & Welling, 2013; Kingma & Dhariwal, 2018; van den Oord et al., 2016); thus, recent works have considered using deep generative models to detect out-of-distribution samples (Li et al., 2018). However, contrary to popular belief, density estimates by deep generative models are highly inaccurate (Nalisnick et al., 2018). For example, a Glow (Kingma & Dhariwal, 2018) model trained on CIFAR10 gives higher likelihood estimates to SVHN samples than CIFAR10 ones, which makes accurate OoD detection impossible. While alternative statistics based on likelihood estimates have been proposed to alleviate this issue (Choi et al., 2018; Song et al., 2017), they are not able to detect OoD samples consistently.

In this paper, we propose a new statistic to detect OoD samples using deep generative models trained with batch normalization (Ioffe & Szegedy, 2015). We argue that using batch normalization not only improves optimization (as argued in (Kohler et al., 2018; Santurkar et al., 2018)), but also challenges the i.i.d. assumption underlying typical likelihood-based objectives. We show in that the training objective of generative models with batch normalization can be interpreted as maximum pseudo-likelihood over a different joint distribution that does not assume data in the same batch are i.i.d. samples. Empirically, we demonstrate that *over this joint distribution, the estimated likelihood of a batch of OoD samples is much lower than that of in-distribution samples*. This allows us to propose a permutation test which outperforms existing methods by a significant margin without modifying how the underlying generative model is trained. In particular, we achieve near-perfect performance even on cases such as Fashion MNIST vs. KMNIST, where the likelihood distributions for single samples are nearly identical (see Figure 1, left). While generative models trained with BatchNorm

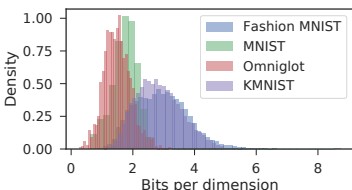 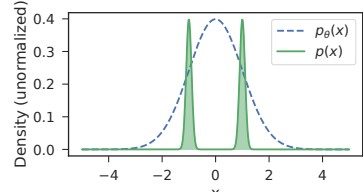

Figure 1: (Left) Density estimation with a RealNVP model trained on Fashion MNIST. The model assigns similar / higher likelihoods to several OoD datasets. (Right) Model mis-specification can result in OoD samples having higher log-likelihoods.

might not provide state-of-the-art likelihood numbers on the test set, it remains competitive and could be more suited for the OoD detection problem (Kingma & Dhariwal, 2018; Nalisnick et al., 2018).

## 2 BACKGROUND

**Setup** Let $\boldsymbol{X} = \{\boldsymbol{x}_i\}_{i=1}^N$ be a set of $N$ observations; we assume that "in-distribution" samples are drawn i.i.d. from some underlying distribution $\boldsymbol{x} \sim p(\boldsymbol{x})$. Our goal is to detect whether a test sample $\boldsymbol{x}$ is "out-of-distribution"; we consider drawing the test samples from an unknown distribution $q(\boldsymbol{x})$, but we do not have access to samples from $q(\boldsymbol{x})$ during training. Following definitions in (Hendrycks & Gimpel, 2016; Liang et al., 2017), we define OoD examples as the test samples that have low densities under $p(\boldsymbol{x})$. We note that samples from $q(\boldsymbol{x})$ are not necessarily all OoD; for example, if $q = p$ then most samples should not be classified as OoD. Detection of OoD samples is useful for applications such as anomaly detection (Li et al., 2018) and exploration in reinforcement learning (Ostrovski et al., 2017).

**Likelihood-based generative models for out-of-distribution detection** Likelihood-based models are trained by finding parameters $\boldsymbol{\theta}$ such that the corresponding $p_{\boldsymbol{\theta}}(\boldsymbol{x})$ is as close as possible to the training distribution $p(\boldsymbol{x})$, in terms of Kullback-Leibler divergence (KL):

$$\arg\min_{\boldsymbol{\theta}} D_{\mathrm{KL}}(p(\boldsymbol{x})\|p_{\boldsymbol{\theta}}(\boldsymbol{x})) = \arg\min_{\boldsymbol{\theta}} \mathbb{E}_{p(\boldsymbol{x})}\left[\log p(\boldsymbol{x}) - \log p_{\boldsymbol{\theta}}(\boldsymbol{x})\right]$$

Finding appropriate solutions depends heavily on choosing the right family of (parametrized) probability distributions. In the context of images, deep neural networks have demonstrated good performance. Likelihood-based deep generative models include variational autoencoders (Kingma & Welling, 2013) (where variational lower bounds are optimized), autoregressive models (van den Oord et al., 2016; Salimans et al., 2017), and invertible flow models (Dinh et al., 2016; Kingma & Dhariwal, 2018) (where exact likelihood is optimized).

Given a sample $\boldsymbol{x}$, we could attempt to test whether it is OoD or not by evaluating its likelihood $-\log p_{\boldsymbol{\theta}}(\boldsymbol{x})$. Intuitively, if $p_{\boldsymbol{\theta}}(\boldsymbol{x})$ is sufficiently similar to $p(\boldsymbol{x})$, OoD samples should have low likelihood under $p_{\boldsymbol{\theta}}(\boldsymbol{x})$. However, this is not the case for some deep generative models (Nalisnick et al., 2018; Choi et al., 2018), where OoD samples are observed to have higher likelihoods than samples from the training distribution $p(\boldsymbol{x})$ across several datasets and models. For example, state-of-the-art models trained on CIFAR10 (Krizhevsky & Hinton, 2009) assign higher likelihood to SVHN (Netzer et al., 2011) samples than CIFAR10 samples. A similar phenomenon has also been observed by Nalisnick et al. (2018) for models trained on Fashion MNIST (Xiao et al., 2017) (Figure 1 (left)), where images with lower variance across pixels (such as MNIST) are assigned higher likelihoods. This invalidates the assumption that OoD samples are assigned lower likelihoods by existing deep generative models.

To alleviate this issue, Choi et al. (2018) use Watanabe Akaike Information Criterion (WAIC):

$$-\mathrm{WAIC}(\boldsymbol{x}) = -\mathbb{E}_{\boldsymbol{\theta}}[\log p_{\boldsymbol{\theta}}(\boldsymbol{x})] + \mathrm{Var}_{\boldsymbol{\theta}}[\log p_{\boldsymbol{\theta}}(\boldsymbol{x})] \tag{1}$$

where the expectations are computed using independently trained model ensembles. This method does not assign higher negative WAIC values for OoD samples in some simple cases, but is empirically observed it to be effective for certain datasets and generative models.

**Likelihood-based permutation tests**   In the context of detecting adversarial examples, Song et al. (2017) considered using permutation tests statistics to determine whether an input $\boldsymbol{x}'$ comes from $p(\boldsymbol{x})$ or not. One such test statistic uses the rank of $p_{\boldsymbol{\theta}}(\boldsymbol{x}')$ in $\{p_{\boldsymbol{\theta}}(\boldsymbol{x}_1), \cdots, p_{\boldsymbol{\theta}}(\boldsymbol{x}_N)\}$, where $\{\boldsymbol{x}_i\}_{i=1}^N$ is a (training) dataset of $N$ samples from $p(\boldsymbol{x})$:

$$T_{\mathrm{perm}} = T(\boldsymbol{x}'; \boldsymbol{x}_1, \ldots, \boldsymbol{x}_N) \triangleq \left| \sum_{i=1}^N \mathbb{I}[p_{\boldsymbol{\theta}}(\boldsymbol{x}_i) \leq p_{\boldsymbol{\theta}}(\boldsymbol{x}')] - \frac{N}{2} \right| \in [0, N/2 + 1] \qquad (2)$$

where $\mathbb{I}$ is the indicator function. When $T_{\mathrm{perm}}(\boldsymbol{x})$ is large, $\boldsymbol{x}$ is assumed to be OoD. This addresses the case where higher likelihood is observed for certain out-of-distribution samples (such as the SVHN vs. CIFAR10 example in Nalisnick et al. (2018)), as both "high-likelihood" and "low-likelihood" samples will have $T_{\mathrm{perm}}$ close to $N/2$ (because of the absolute value) and be identified as OoD.

## 3   OUT-OF-DISTRIBUTION SAMPLES AND MODEL MIS-SPECIFICATION

Nalisnick et al. (2018) find it surprising that deep generative models assign higher likelihood to out-of-distribution samples, given their successful generalizing on a test dataset. However, we argue that this is to be expected when the unerlying generative model $p_\theta(\boldsymbol{x})$ is mis-specified, and perhaps this is more likely than previously anticipated in the literature.

We consider a simple example where a mis-specified model would overestimate the likelihood of out-of-distribution samples (Figure 1, right). Let $p(\boldsymbol{x})$ be a uniform mixture of two Gaussians on $\mathbb{R}^1$, $\mathcal{N}(-1.0, 0.01)$ and $\mathcal{N}(1.0, 0.01)$, and $p_\theta(\boldsymbol{x}) = \mathcal{N}(\mu, \sigma^2)$ be our model with parameters $(\mu, \sigma)$. The maximum likelihood solution is $(\mu^\star, \sigma^\star) \approx (0.0, 1.0)$, where mode-covering behavior occurs. $\boldsymbol{x} = 0$ has the largest likelihood under $p_\theta(\boldsymbol{x})$, yet it is highly atypical in the original distribution. This is also a failure mode for WAIC as discussed in (Choi et al., 2018).

Given the complexity of high-dimensional image distributions, existing likelihood-based generative models are likely to be mis-specified, which invalidates the use of likelihood estimates to perform OoD detection (White, 1982). Moreover, model selection is often based on the holdout method (Arlot et al., 2010), in which we evaluate log-likelihood over a validation set sampled from $p(\boldsymbol{x})$, but not over out-of-distribution samples. As we do not know the entropy of $p(\boldsymbol{x})$, we can never check whether $D_{\mathrm{KL}}(p(\boldsymbol{x})\|p_{\boldsymbol{\theta}}(\boldsymbol{x})) \approx 0$ after training. Because of the mode-seeking nature of $D_{\mathrm{KL}}$, an alternative distribution $q(\boldsymbol{x}) \neq p(\boldsymbol{x})$ can have even lower KL-divergence with $p_{\boldsymbol{\theta}}(\boldsymbol{x})$, i.e., $D_{\mathrm{KL}}(q(\boldsymbol{x})\|p_{\boldsymbol{\theta}}(\boldsymbol{x})) \ll D_{\mathrm{KL}}(p(\boldsymbol{x})\|p_{\boldsymbol{\theta}}(\boldsymbol{x}))$ even if $p_{\boldsymbol{\theta}}(\boldsymbol{x})$ is trained on samples from $p(\boldsymbol{x})$. In Appendix B, we demonstrate that even more sophisticated tests over log-likelihood, such as likelihood-based permutation tests, cannot detect out-of-distribution samples effectively.

## 4   DETECTING OUT-OF-DISTRIBUTION SAMPLES WITH BATCH NORMALIZATION

In the following sections, we argue that by taking advantage of batch normalization (BatchNorm), out-of-distribution samples can be detected with existing deep generative models. For a batch of inputs $\boldsymbol{z} = \{\boldsymbol{z}_i\}_{i=1}^b$ of batch size $b$, batch normalization (Ioffe & Szegedy, 2015) performs normalization over the inputs followed by a parametrized affine transformation:

$$\mathrm{BatchNorm}(\boldsymbol{z}; \gamma, \beta, \epsilon) = \frac{\boldsymbol{z} - \mathbb{E}[\boldsymbol{z}]}{\sqrt{\mathrm{Var}[\boldsymbol{z}] + \epsilon}} \cdot \gamma + \beta$$

where $\gamma, \beta$ are trainable parameters, and $\epsilon$ is a hyperparameter for numerical stability. The mean $\mathbb{E}[\boldsymbol{z}]$ and variance $\mathrm{Var}[\boldsymbol{z}]$ are computed over a single batch in *training mode*, and over the entire training set in *evaluation mode*[1]. For deep generative models using batch normalization, existing literature evaluate the log-likelihood on some validation set in *evaluation mode*. The role of *training mode* is generally perceived to accelerate optimization (Kohler et al., 2018; Santurkar et al., 2018), similar to supervised learning.

---

[1]This is typically done by keeping an exponential moving average over the batch statistics during training.

Table 1: Log-likelihood (measured in bits per dimension) calculated with RealNVP, VAE, Pixel-CNN++ models on MNIST, Fashion MNIST, CIFAR10, and SVHN test sets. We train RealNVP and VAE on FashionMNIST, and train RealNVP and PixelCNN on CIFAR10. We report likelihood results with *training mode*, *evaluation mode*, and their difference ($\Delta$).

(a) RealNVP (trained on Fashion MNIST)

| Evaluated on | BN Mode | BPD | $\Delta$ |
|---|---|---|---|
| Fashion MNIST | *evaluation* | 2.92 | 0.02 |
| | *training* | 2.94 | |
| MNIST | *evaluation* | 1.74 | 7.73 |
| | *training* | 9.47 | |

(b) RealNVP (trained on CIFAR10)

| Evaluated on | BN Mode | BPD | $\Delta$ |
|---|---|---|---|
| CIFAR10 | *evaluation* | 3.48 | 0.03 |
| | *training* | 3.51 | |
| SVHN | *evaluation* | 2.44 | 8.56 |
| | *training* | 11.10 | |

(c) VAE (trained on Fashion MNIST)

| Evaluated on | BN Mode | BPD | $\Delta$ |
|---|---|---|---|
| Fashion MNIST | *evaluation* | 3.19 | 0.01 |
| | *training* | 3.20 | |
| MNIST | *evaluation* | 1.97 | 6.53 |
| | *training* | 8.50 | |

(d) PixelCNN++ (trained on CIFAR10)

| Evaluated on | BN Mode | BPD | $\Delta$ |
|---|---|---|---|
| CIFAR10 | *evaluation* | 3.21 | 0.12 |
| | *training* | 3.33 | |
| SVHN | *evaluation* | 2.16 | 1.88 |
| | *training* | 4.04 | |

## 4.1 TRAINING MODE DECREASES LIKELIHOOD OF OOD BATCHES

*Our central observation is that likelihoods assigned by the model under training mode and evaluation mode are significantly different for OoD samples, but much less so for in-distribution samples*, i.e., when we use BatchNorm statistics computed over a single batch. We evaluate log-likelihood (measured in bits per dimension, BPD[2]) on Fashion MNIST, MNIST, CIFAR and SVHN test sets with several models including RealNVP, VAE and PixelCNN++[3]. Pre-activation batch normalization layers are used within the residual block for RealNVP and PixelCNN++, and before each convolutional layer for VAE. The datasets are evaluated in both *evaluation mode* and *training mode*.

In Table 1 we found that in *evaluation mode* MNIST samples have smaller BPD than Fashion MNIST for models trained on Fashion MNIST; SVHN samples have smaller BPD than CIFAR10 for models trained on CIFAR10, which agrees with the observations in (Nalisnick et al., 2018). However, as we change from *evaluation mode* to *training mode*, the log-likelihood of the in-distribution samples does not change significantly, while that of OoD samples plummets (see $\Delta$ in Table 1). In Appendix C.3, we demonstrate that this sudden decrease in log-likelihood not only happens for samples from other datasets, but also adversarial samples that are able to fool likelihood models and likelihood-based permutation tests. This empirically suggests that simply by using *training mode* during evaluation, we are able to consistently detect a batch of out-of-distribution samples. We further note that no special modifications is made over the training procedure, and no additional OoD datasets are required.

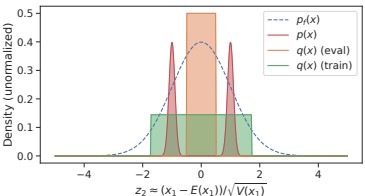

Figure 2: Example demonstrating how BatchNorm mitigates high-likelihood in OoD distributions. $p(x)$, $p_f(x)$ and $q(x)$ denote the original, model and OoD distributions. Distribution of $z_2$ are different for $q(x)$ under the two modes of BatchNorm, leading to different likelihood results.

---

[2]BPD is defined by the negative log-likelihood divided by the number of dimensions (Theis et al., 2015); if the likelihood is measured in nats, then an additional division by $\ln 2$ is needed.

[3]We use an implementation where we replace weight normalization with batch normalization for PixelCNN++, see https://github.com/pclucas14/pixel-cnn-pp.

## 4.2 EXPLAINING THE EFFECTIVENESS OF BATCH NORMALIZATION

Why is batch normalization effective at detecting OoD batches while other likelihood-based generative models (e.g., Glow trained with ActNorm) fail? First we provide an example to illustrate why *training mode* and *evaluation mode* provide different likelihood estimates for OoD distributions. Then we argue that with *training mode*, the learned generative model no longer assumes samples are i.i.d., which differs from models without BatchNorm.

### 4.2.1 EXAMPLE OF BATCHNORM MITIGATING MODEL MIS-SPECIFICATION

In Figure 2, we illustrate a case where BatchNorm could be useful for OoD detection. Suppose we try to learn some distribution $p(\boldsymbol{x})$ with a 2-d flow model with one coupling layer (Dinh et al., 2016): $f : (x_1, x_2) \mapsto (z_1, z_2)$ where $z_1 = x_1$, $z_2 = x_2 + (x_1 - E(x_1))/\sqrt{V(x_1)} \cdot \gamma + \beta$ with learnable parameters $\gamma, \beta$; $E(x_1)$, $V(x_1)$ are the empirical mean and variance of a batch of $x_1$ in *training mode*, and the mean and variance of $p(x_1)$ in *evaluation mode*. To simplify our analysis, We assume that the training distribution $p(x_1, x_2) = p(x_1)p(x_2)$ where $p(x_1)$ is a mixture of two Gaussians ($\mathcal{N}(-1, \sigma^2)$ and $\mathcal{N}(1, \sigma^2)$ with $\sigma \to 0$) and $p(x_2) = \delta_0(x_2)$ (Dirac delta) and batch sizes are infinite; hence $z_2 \approx (x_1 - E(x_1))/\sqrt{V(x_1)} \cdot \gamma + \beta$. Since the Jacobian of $f$ is triangular and has a determinant of 1, the likelihood of the flow model is simply $p_f(x_1, x_2) = p(f(x_1, x_2))$; we assume the prior $p(z_1, z_2)$ is standard Gaussian $\mathcal{N}(0, I)$. The optimal parameters are then $\gamma \approx 1^4$ and $\beta = 0$.

Now let us consider $q(x_1, x_2) = q(x_1)p(x_2)$ where $q(x_1) = \text{Uniform}(-0.5, 0.5)$. In *evaluation mode*, $z_2 \approx x_1$ and $\mathbb{E}_q[\log p_f(\boldsymbol{x})] \approx -1.92$; in *training mode*, $z_2 \approx \sqrt{12}x_1$, which decreases the log-likelihood to $\mathbb{E}_q[\log p_f(\boldsymbol{x})] \approx -2.38$. We include a more detailed analysis in Appendix D. In this case, BatchNorm manages to learn a global property of the input distribution (variance), and use this to counteract major distributional shifts (e.g. the proposed uniform distribution $q(x)$). The effect leads to lower log-likelihoods in *training mode* compared to that in *evaluation mode*.

### 4.2.2 REVISITING THE I.I.D. ASSUMPTIONS FOR BATCH-NORMALIZED GENERATIVE MODELS

While it is difficult to analytically characterize the change of likelihood estimates in *training mode* for high-dimensional images and deep neural networks, we provide a probabilistic interpretation as to why *training mode* exhibits different behaviors than *evaluation mode*.

***Training mode*** **breaks the *i.i.d.* data assumption**     In *training mode*, the output for a particular sample $\boldsymbol{x}_j$ is affected by other samples in the same batch (denoted as $\boldsymbol{x}_{-j}$), so they are not treated as i.i.d for the generative models in *training mode*. This suggest that using BatchNorm when training generative models not only changes the optimization landscape but also modifies the objective: it is no longer $\frac{1}{N} \sum_{i=1}^{N} \log p_{\boldsymbol{\theta}}(\boldsymbol{x}_i)$ where each sample is assumed to be i.i.d.

**Pseudo-likelihood perspective of *training mode***     How should we interpret the training objective with batch normalization? For each sample $\boldsymbol{x}_j$, its corresponding "likelihood" objective depends on *the other samples from the same batch*; we denote this as $\ell_{\boldsymbol{\theta}}(\boldsymbol{x}_j; \boldsymbol{x}_{-j})$. First, we show that the integral of $\exp(\ell_{\boldsymbol{\theta}}(\boldsymbol{x}_j; \boldsymbol{x}_{-j}))$ over any particular sample ($\boldsymbol{x}_j$) when we fix the other samples in the batch ($\boldsymbol{x}_{-j}$) is one; this allows us to treat $\exp(\ell)$ as a probability density function[5].

**Proposition 1.** *Let $b \in \mathbb{N}, b > 1$ be the batch size, and for all possible batch of samples of size $(b - 1)$, denoted as $\boldsymbol{x}_{-j}$, if $\ell_{\boldsymbol{\theta}}(\boldsymbol{x}_j; \boldsymbol{x}_{-j})$ is the training objective over $\boldsymbol{x}_j$ for a likelihood-based generative model with batch normalization, then:*

$$\int_{\boldsymbol{x}_j} \exp(\ell_{\boldsymbol{\theta}}(\boldsymbol{x}_j; \boldsymbol{x}_{-j})) \mathrm{d}\boldsymbol{x}_j = 1.$$

*where we consider likelihood-based generative models that sample either via some parametrized distribution (such as VAE and PixelCNN) or via parametrized invertible transformations constructed via affine coupling layers (such as RealNVP).*

---

[4]By maximum likelihood, $\gamma \approx \arg\min_s -s^2/2 + \log s = 1$.

[5]For VAE, we assume that the inference distribution is always optimal, i.e., there is no gap between the evidence lower bound and the log-likelihood, so we can treat the objective as exact maximum likelihood.

We defer the proofs in Appendix A.1. Moreover, if we interpret $\ell_\theta(\boldsymbol{x}_j; \boldsymbol{x}_{-j})$ as the *conditional* log-likelihood under a certain joint generative model $\tilde{p}_\theta$ over an entire batch, our existing "log-likelihood" objective is a surrogate to the log-likelihood objective over the joint distribution $\tilde{p}_\theta(\boldsymbol{x}_1, \boldsymbol{x}_2, \ldots, \boldsymbol{x}_b)$.

**Proposition 2.** *There exists a joint distribution $\tilde{p}_\theta(\boldsymbol{x}_1, \boldsymbol{x}_2, \ldots, \boldsymbol{x}_b)$ such that for all $j$, $\tilde{p}_\theta(\boldsymbol{x}_j | \boldsymbol{x}_{-j}) \to \ell_\theta(\boldsymbol{x}_j; \boldsymbol{x}_{-j})$ as $b \to \infty$. Then, the objective for one batch $\{x_j\}_{j=1}^b$ in training mode*

$$\mathcal{L}_{train}(\{x_j\}_{j=1}^b; \boldsymbol{\theta}) = \sum_{j=1}^b \ell_\theta(\boldsymbol{x}_j | \boldsymbol{x}_{-j}) \triangleq \sum_{j=1}^b \log \tilde{p}_\theta(\boldsymbol{x}_j | \boldsymbol{x}_{-j}) \tag{3}$$

*is the pseudo-log-likelihood for the joint distribution $\tilde{p}_\theta(\boldsymbol{x}_1, \boldsymbol{x}_2, \ldots, \boldsymbol{x}_b)$ as $b \to \infty$.*

***Training mode* versus *evaluation mode*** In *evaluation mode*, the samples are treated as i.i.d., since the batch statistics is fixed and samples within the same batch do not affect each other. The objective for one batch $\{x_j\}_{j=1}^b$ evaluated in *evaluation mode* is simply:

$$\mathcal{L}_{\text{eval}}(\{x_j\}_{j=1}^b; \boldsymbol{\theta}) = \sum_{j=1}^b \log p_\theta(\boldsymbol{x}_j) \tag{4}$$

It is evident that $\mathcal{L}_{\text{train}}$ and $\mathcal{L}_{\text{eval}}$ are not the same objective due to the differences in i.i.d. assumptions for samples within the same batch. $\mathcal{L}_{\text{train}}$ depends on the input batch statistics, whereas $\mathcal{L}_{\text{eval}}$ does not. If we consider OoD batches that exhibit major distributional shifts, then it is likely that $\mathcal{L}_{\text{train}}$ differs from $\mathcal{L}_{\text{eval}}$. From the empirical evidence in Table 1, OoD samples have much lower BPD on *training mode* than on *evaluation mode*, whereas the same does not happen for in-distribution samples.

## 5 A PERMUTATION TEST BASED ON BATCH NORMALIZATION

Assume that during testing we have access to some "test dataset" $\hat{\boldsymbol{X}}$, and our goal is to identify for each test sample $\hat{\boldsymbol{x}} \in \hat{\boldsymbol{X}}$ whether it is OoD or not. Based on the observations over OoD batches, we propose a permutation test for each test sample $\hat{\boldsymbol{x}}$.

**Interpolation between *training* and *evaluation*** First, we propose a procedure that interpolates between *training mode* and *evaluation mode*, generalizing the settings in Section 4. We randomly obtain a batch of samples $\hat{\boldsymbol{X}}_{b_1}$ from $\hat{\boldsymbol{X}}$ with size $b_1$ and a batch of samples $\boldsymbol{X}_{b_0}$ from $p(\boldsymbol{x})$ with size $b_0$ (e.g., from the training dataset $\boldsymbol{X}$ used to fit the generative model). For each $\hat{\boldsymbol{x}}_j \in \hat{\boldsymbol{X}}_{b_1}$ indexed by $j$, we compute its log-likelihood under *training mode* by mixing $\hat{\boldsymbol{X}}_{b_1}$ and $\boldsymbol{X}_{b_0}$ in the same batch:

$$\log \tilde{p}_\theta(\hat{\boldsymbol{x}}_j | \boldsymbol{x}_{-j}) \quad \text{where} \quad \boldsymbol{x}_{-j} = (\boldsymbol{X}_{b_0} \cup \hat{\boldsymbol{X}}_{b_1}) \setminus \{\hat{\boldsymbol{x}}_j\} \tag{5}$$

Let $b = b_1 + b_0$ be the batch size and let $r = b_1/b$ be the ratio of the test samples in the entire batch. We note that the above procedure interpolates evaluation with *training mode* and evaluation with *evaluation mode*. As $r \to 0$, the effect on the batch statistics is dominated by the training samples $\boldsymbol{X}_{b_0}$, so Equation 5 converges to *evaluation mode*; as $r \to 1$, the effect on the batch statistics is dominated by the test samples $\hat{\boldsymbol{X}}_{b_1}$, so Equation 5 converges to *training mode*. As we increase $r$ from 0 to 1, we are moving from *evaluation mode* to *training mode*.

The observations in Section 4 suggest that $\log \tilde{p}_\theta(\hat{\boldsymbol{x}}_j | \boldsymbol{x}_{-j})$ should be relatively stable if $q(\boldsymbol{x})$ is close to $p(\boldsymbol{x})$ and increase significantly if $q(\boldsymbol{x})$ is far from $p(\boldsymbol{x})$, which is indeed the case empirically. We consider evaluating the average log-likelihood on the test samples when the ratio $r$ is varied, and show the results on several datasets and models in Figure 3. As we increase the ratio of test samples, the corresponding log-likelihood decreases dramatically for OoD samples, but is relatively stable for in-distribution samples.

**Permutation test with BatchNorm** We proceed to propose a permutation test statistics based on *training mode* evaluation. For a test sample $\hat{x}$, a fixed batch size $b$ and fixed ratio $r$, we select $(rb - 1)$ samples from $q(\boldsymbol{x})$ and $(1 - r)b$ samples from $p(\boldsymbol{x})$. The expected conditional likelihood of $\hat{x}$ is used as the test statistic:

$$S_{b,r}(\boldsymbol{x}') = \mathbb{E}_{p,q} \left[ \log \tilde{p}_\theta(\hat{\boldsymbol{x}} | \boldsymbol{X}_{(1-r)b}, \hat{\boldsymbol{X}}_{rb-1}) \right] \quad \text{where} \quad \boldsymbol{X}_{(1-r)b} \sim p(\boldsymbol{x}), \hat{\boldsymbol{X}}_{rb-1} \sim q(\boldsymbol{x}) \tag{6}$$

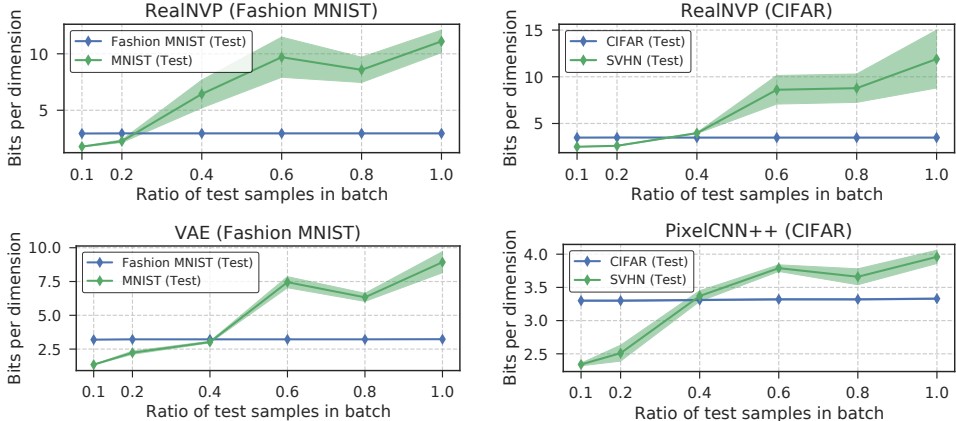

Figure 3: Average BPD ($-\log \tilde{p}_{\boldsymbol{\theta}}(\boldsymbol{x}_j|\boldsymbol{x}_{-j})$) for test samples with varying ratios of test samples in the batch. The BPD of in-distribution samples do not increase as the ratio increase, yet that of OoD samples increase significantly. This justifies the use of $\Delta_{b,r_1,r_2}(\boldsymbol{x}')$ for OoD detection.

In practice, we use monte carlo estimates for the expectation in $S_{b,r}(\boldsymbol{x}')$. We select two different ratios $r_1, r_2 \in (0,1)$ where $r_1 < r_2$, and compute the differences in $S_{b,r}(\boldsymbol{x}')$:

$$\Delta_{b,r_1,r_2}(\boldsymbol{x}') = |S_{b,r1}(\boldsymbol{x}') - S_{b,r2}(\boldsymbol{x}')| \tag{7}$$

We proceed to use the rank of $\Delta_{b,r_1,r_2}(\hat{\boldsymbol{x}})$ in the training set as our test statistic:

$$T_{b,r_1,r_2} = T(\boldsymbol{x}'; \boldsymbol{x}_1, \ldots, \boldsymbol{x}_N) \triangleq \sum_{i=1}^{N} \mathbb{I}[\Delta_{b,r_1,r_2}(\boldsymbol{x}_i) \leq \Delta_{b,r_1,r_2}(\boldsymbol{x}')] \tag{8}$$

For in-distribution samples, we expect $\Delta_{b,r_1,r_2}$ to be small across all choices of $r_1$ and $r_2$; for out-of-distribution samples, $\Delta_{b,r_1,r_2}$ should be large if $(r_2 - r_1)$ is large. Therefore, if $q(\boldsymbol{x}) = p(\boldsymbol{x})$, then the statistics computed over the $\boldsymbol{x} \sim q(\boldsymbol{x})$ should be approximately uniformly distributed over $[0, N]$, whereas if $q(\boldsymbol{x}) \neq p(\boldsymbol{x})$, then the statistics computed over out-of-distribution samples would be concentrated around $N$.

## 6 EXPERIMENTS

We verify the effectiveness of our proposed test statistic on several datasets and models, including RealNVP, VAE and PixelCNN++, and compare against several baselines including log-likelihood, permutation test and WAIC. First, we train a model $\tilde{p}_{\boldsymbol{\theta}}(\boldsymbol{x})$ on a dataset $p(\boldsymbol{x})$; then we proceed to obtain the test statistic for each sample in $q(\boldsymbol{x})$, where $q(\boldsymbol{x})$ are several different datasets. For each $q(\boldsymbol{x}) \neq p(\boldsymbol{x})$, we consider a binary classification problem, where the prediction is the test statistic for each sample $\boldsymbol{x}$, and the label is whether $\boldsymbol{x}$ is out-of-distribution (label 1) or not (label 0); larger prediction values indicate $\boldsymbol{x}$ is more likely to be out-of-distribution. We evaluate the area under the ROC curve (AUC) and average precision (AP) for each binary classification task, following the procedure in (Hendrycks & Gimpel, 2016). We select $r_1 = 0.1$ and $r_2 = 0.9$ for our proposed test statistic, and the batch size $B = 64$.

The OoD detection results are shown in Table 2. As expected, using $\log p_{\boldsymbol{\theta}}(\boldsymbol{x})$ results in poor AUC and AP in cases where the OoD samples have higher log-likelihoods; $T_{\text{perm}}$ outperforms $\log p_{\boldsymbol{\theta}}(\boldsymbol{x})$ in cases where we also consider high log-likelihood samples as OoD; WAIC achieves higher AUC / AP than $\log p_{\boldsymbol{\theta}}(\boldsymbol{x})$, yet its improvement is inconsistent across different models (RealNVP improvements are lower than VAE on Fashion MNIST). Our proposed statistic is able to detect all the out-of-distribution samples by achieving near-optimal AUC / AP in most cases; this is most notable on CIFAR vs. ImageNet and Fashion MNIST vs. KMNIST as the sample have very similar likelihood distributions in *evaluation mode*. In Appendix C.4, we show that our method works well even with $r_1 \to 0$ and $r_2 = 0.15$; therefore, the method is not very sensitive to the $r_1, r_2$ hyperparameters. This

Table 2: Out-of-distribution classification evaluated with AUC (left) and Average Precision (right). Higher is better. Rotation denotes $q(\boldsymbol{x})$ uses images in $p(\boldsymbol{x})$ yet each image is rotated randomly by $d \in (90, 270)$ degrees.

| $p(\boldsymbol{x})$ | Model | $q(\boldsymbol{x})$ | $\log p_{\boldsymbol{\theta}}(\boldsymbol{x})$ | $T_{\mathrm{perm}}$ | WAIC | Ours |
|---|---|---|---|---|---|---|
| Fashion MNIST | RealNVP | Rotation | 0.76 / 0.78 | 0.64 / 0.68 | **0.99 / 0.99** | **0.99 / 0.99** |
| | | MNIST | 0.10 / 0.32 | 0.78 / 0.71 | 0.24 / 0.38 | **1.00 / 1.00** |
| | | Omniglot | 0.05 / 0.06 | 0.86 / 0.80 | 0.97 / 0.95 | **1.00 / 1.00** |
| | | KMNIST | 0.47 / 0.46 | 0.45 / 0.45 | 0.63 / 0.65 | **1.00 / 1.00** |
| Fashion MNIST | VAE | Rotation | 0.73 / 0.72 | 0.61 / 0.64 | 0.94 / 0.95 | **0.97 / 0.98** |
| | | MNIST | 0.13 / 0.33 | 0.73 / 0.68 | 0.56 / 0.64 | **1.00 / 1.00** |
| | | Omniglot | 0.00 / 0.06 | 0.99 / 0.96 | 0.90 / 0.83 | **1.00 / 1.00** |
| | | KMNIST | 0.55 / 0.54 | 0.50 / 0.50 | 0.84 / 0.87 | **1.00 / 1.00** |
| CIFAR10 | RealNVP | Rotation | 0.87 / 0.87 | 0.79 / 0.79 | 0.99 / 0.98 | **1.00 / 1.00** |
| | | SVHN | 0.07 / 0.52 | 0.86 / 0.82 | 0.16 / 0.55 | **1.00 / 1.00** |
| | | ImageNet | 0.51 / 0.52 | 0.50 / 0.51 | 0.58 / 0.59 | **0.98 / 0.97** |
| | | LSUN | 0.70 / 0.39 | 0.58 / 0.56 | 0.60 / 0.28 | **0.99 / 0.98** |
| | | CIFAR100 | 0.52 / 0.54 | 0.55 / 0.55 | 0.53 / 0.54 | **0.71 / 0.75** |
| CIFAR10 | PixelCNN++ | Rotation | 0.77 / 0.75 | 0.67 / 0.63 | 0.90 / 0.85 | **0.99 / 0.99** |
| | | SVHN | 0.10 / 0.32 | 0.86 / 0.77 | 0.09 / 0.53 | **0.99 / 0.99** |
| | | ImageNet | 0.51 / 0.51 | 0.49 / 0.50 | 0.66 / 0.69 | **0.89 / 0.87** |
| | | LSUN | 0.72 / 0.69 | 0.60 / 0.58 | 0.78 / 0.74 | **0.98 / 0.97** |

suggest that in practice, even when the test samples contains only a small portion of the OoD samples we can still detect them reliably by selecting $r_1 \to 0$ (evaluation mode) and $r_2 \to 1$ (training mode).

Additionally, we also perform permutation tests over entire batches of test samples, where the BPD of *evaluation mode* are averaged. The AUC / AP of CIFAR10 versus OoD datasets are 1.00 / 1.00 (SVHN), 0.56 / 0.64 (CIFAR100), 0.82 / 0.68 (LSUN), and 0.56 / 0.68 (ImageNet). These results are significantly lower than evaluated with *training mode* where the AUC / AP are near-perfect, which demonstrates that utilizing *training mode* is behind the improvements for OoD detection.

# 7 RELATED WORK

**Task-dependent OoD detection**  Out-of-distribution detection is crucial to applications such as anomaly detection (Pidhorskyi et al., 2018; Hendrycks & Gimpel, 2016; Vyas et al., 2018), adversarial defense (Song et al., 2017; 2018), and novelty detection for exploration (Marsland, 2003; Bellemare et al., 2016; Fu et al., 2017). In the context of supervised learning, OoD detection methods (Hendrycks & Gimpel, 2016; Liang et al., 2017; DeVries & Taylor, 2018; Gal & Ghahramani, 2016; Lakshminarayanan et al., 2017) are applied to prevent poorly-calibrated neural networks (Guo et al., 2017) from making high-confidence predictions on nonsensical inputs (Szegedy et al., 2013; Goodfellow et al., 2014); these methods are task dependent and are not suitable for task-independent cases, such as exploration over novel states (Fu et al., 2017; Ostrovski et al., 2017).

**Generative models for task-independent OoD detection**  For high-dimensional inputs such as images, deep generative models are widely applied for density estimation (van den Oord et al., 2016; Dinh et al., 2016; Kingma & Dhariwal, 2018), and thus are naturally considered for task-independent OoD detection (Chalapathy et al., 2018; Xu et al., 2018; Kliger & Fleishman, 2018; Ostrovski et al., 2017; Li et al., 2018). However, recent empirical evidence suggest that likelihood estimates by popular deep generative models are not reliable enough for OoD detection, even against samples that are not adversarial by construction (Nalisnick et al., 2018; Hendrycks et al., 2018). To address these issues, Škvára et al. propose tuning the hyperparameters of VAEs with additional OoD data. Choi et al. (Choi et al., 2018) address this empirically by assuming that OoD samples have higher variance likelihood estimates under different independently trained models. Ren et al. consider learning a separate density model for potentially confounding background statistics.

**Statistical tests for OoD detection**   OoD detection can be posed as a hypothesis test where the null hypothesis assumes the data is OoD. One could also consider a more specific goodness-of-fit (GoF) test which determines whether a dataset is drawn from a specific distribution. Nalisnick et al. (2019) considers Kernelized Stein discrepancy (Chwialkowski et al., 2016; Liu et al., 2016) and Maximum-mean discrepancy (Gretton et al., 2012) as two GoF tests that scales to generative models. They propose to compare the empirical cross-entropy of the samples with entropy of the dataset, which is related to the permutation test proposed in Song et al. (2017). Our permutation test can also be cast as a GoF test with $r \to 1$, where all test samples in the same batch with *training mode*, but we can also provide individual OoD predictions via different $r$ values.

## 8   CONCLUSION

In this work, we revisit the i.i.d. assumptions in generative models trained with batch normalization, which results in an alternative interpretation to the training objective. The difference between the training objective and the evaluation objective explains the intriguing observation that likelihood estimates over OoD examples in *training mode* are much lower than that in *evaluation mode*. This allows us to develop a permutation test based on batch normalization, with which we can reliably detect OoD examples even for difficult cases such as Fashion MNIST vs. KMNIST.

We argue that batch normalization is merely one approach to introducing non-independence between observation variables in deep generative models. Neural processes (Garnelo et al., 2018), for example, considers generative models conditioned on certain observation contexts, which may differ significantly between regular and OoD batches. It would be interesting to investigate if such models could be utilized for more reliable out-of-distribution detection methods for high-dimensional data.

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

## A PROOFS

### A.1 PROOF OF PROPOSITION 1

**Proposition 1.** *Let $b \in \mathbb{N}, b > 1$ be the batch size, and for all possible batch of samples of size $(b-1)$, denoted as $\boldsymbol{x}_{-j}$, if $\ell_\theta(\boldsymbol{x}_j; \boldsymbol{x}_{-j})$ is the training objective over $\boldsymbol{x}_j$ for a likelihood-based generative model with batch normalization, then:*

$$\int_{\boldsymbol{x}_j} \exp(\ell_{\boldsymbol{\theta}}(\boldsymbol{x}_j; \boldsymbol{x}_{-j})) \mathrm{d}\boldsymbol{x}_j = 1.$$

*where we consider likelihood-based generative models that sample either via some parametrized distribution (such as VAE and PixelCNN) or via parametrized invertible transformations constructed via affine coupling layers (such as RealNVP).*

*Proof.* Note that in *training mode*, we can write down the batch norm function explicitly:

$$\mathrm{BatchNorm}(\boldsymbol{z}; \gamma, \beta, \epsilon) = \frac{\boldsymbol{z} - \frac{1}{b}\sum_{i=1}^{b} \boldsymbol{z}_i}{\sqrt{\frac{1}{b-1}\sum_{i=1}^{b}(\boldsymbol{z}_i - \frac{1}{b}\sum_{i=1}^{b}\boldsymbol{z})^2 + \epsilon}} \cdot \gamma + \beta$$

if we fix $\boldsymbol{z}_{-i}$, $\gamma$, $\beta$ and $\epsilon$, then $\mathrm{BatchNorm}(\boldsymbol{z}; \gamma, \beta, \epsilon)$ is a deterministic function over $\boldsymbol{z}_i$. The corresponding neural network $g_\theta(\boldsymbol{x}_i; \boldsymbol{x}_{-i})$ is also a deterministic function over $\boldsymbol{x}_i$ (given $\boldsymbol{x}_{-i}$).

We summarize likelihood-based generative models into two types:

- In the first type, the neural network produces the parameters for a certain tractable distribution, which is used to compute the likelihood (or its lower bound). This includes VAE and autoregressive models. Since the distribution over $\boldsymbol{x}_i$ is parametrized by the deterministic function $g_\theta$, so the likelihood is naturally normalized.

- In the second type, RealNVP models are composed by affine coupling layers (Dinh et al., 2014), which contains transformations $(x, y) \mapsto (x', y')$ where

$$x' = x, \quad y' = s(x) \cdot y + t(x) \tag{9}$$

where $s(\cdot)$ and $t(\cdot)$ are neural networks (potentially with batch normalization layers). This has the reverse

$$x = x', \quad y = (y' - t(x'))/s(x') \tag{10}$$

Since BatchNorm is a deterministic function over $\boldsymbol{z}_i$ when $\boldsymbol{z}_{-i}$ is fixed, the corresponding $s(\cdot)$ and $t(\cdot)$ functions also have the same property. Therefore, the corresponding RealNVP model over $\boldsymbol{x}_i$ with fixed $\boldsymbol{x}_{-i}$ is also invertible.

As $g_\theta$ is a deterministic function over $\boldsymbol{x}_i$, the corresponding conditional likelihood is also normalized.

This completes the argument for likelihood-based generative models. □

### A.2 PROOF OF PROPOSITION 2

**Proposition 2.** *There exists a joint distribution $\tilde{p}_{\boldsymbol{\theta}}(\boldsymbol{x}_1, \boldsymbol{x}_2, \dots, \boldsymbol{x}_b)$ such that for all $j$, $\tilde{p}_{\boldsymbol{\theta}}(\boldsymbol{x}_j | \boldsymbol{x}_{-j}) \to \ell_{\boldsymbol{\theta}}(\boldsymbol{x}_j; \boldsymbol{x}_{-j})$ as $b \to \infty$. Then, the objective for one batch $\{x_j\}_{j=1}^{b}$ in training mode*

$$\mathcal{L}_{train}(\{x_j\}_{j=1}^{b}; \boldsymbol{\theta}) = \sum_{j=1}^{b} \ell_{\boldsymbol{\theta}}(\boldsymbol{x}_j | \boldsymbol{x}_{-j}) \triangleq \sum_{j=1}^{b} \log \tilde{p}_{\boldsymbol{\theta}}(\boldsymbol{x}_j | \boldsymbol{x}_{-j}) \tag{3}$$

*is the pseudo-log-likelihood for the joint distribution $\tilde{p}_{\boldsymbol{\theta}}(\boldsymbol{x}_1, \boldsymbol{x}_2, \dots, \boldsymbol{x}_b)$ as $b \to \infty$.*

*Proof.* We treat each sample in the batch as a separate random variable, and consider the corresponding graphical model where batch normalization is involved. Since *training mode* aggregates the statistics of all the samples, all the random variables are connected to each other.

First, we need to show that for large enough $b$, there exists a joint distribution $\tilde{p}_{\boldsymbol{\theta}}(\boldsymbol{x}_1, \boldsymbol{x}_2, \ldots, \boldsymbol{x}_b)$ that is compatible with all the conditional distributions we defined (Arnold & Press, S. James, 1989). From Theorem 4.1 in (Arnold & Press, S. James, 1989), we need to show that there exists a function $f$ such that

$$\frac{\tilde{p}_{\boldsymbol{\theta}}(\boldsymbol{x}_i|\boldsymbol{x}_{-i})}{\tilde{p}_{\boldsymbol{\theta}}(\boldsymbol{x}_j|\boldsymbol{x}_{-j})} = \frac{f(\boldsymbol{x}_{-j})}{f(\boldsymbol{x}_{-i})} \tag{11}$$

Let $f(\boldsymbol{x}_{-j}) = \mathbb{E}_{\boldsymbol{x}_j}[\sum_{k \neq j} \log \tilde{p}_{\boldsymbol{\theta}}(\boldsymbol{x}_k|\boldsymbol{x}_{-k})]$ be the sum of conditional log-likelihood of $\boldsymbol{x}_k$ under the expectation of some distribution over $\boldsymbol{x}_j$. For large enough $b$, the value of $\boldsymbol{x}_j$ will not affect the batch statistics, so $f(\boldsymbol{x}_{-j}) = \sum_{k \neq j} \log \tilde{p}_{\boldsymbol{\theta}}(\boldsymbol{x}_k|\boldsymbol{x}_{-k})$, so the condition for compatibility in Equation 11 holds, and there exists a compatible joint distribution, which we denote as $\tilde{p}_{\boldsymbol{\theta}}(\boldsymbol{x}_1, \boldsymbol{x}_2, \ldots, \boldsymbol{x}_b)$.

Next, we verify the conditions under which the product of the conditional likelihoods is a pseudo-likelihood of $\tilde{p}_{\boldsymbol{\theta}}(\boldsymbol{x}_1, \boldsymbol{x}_2, \ldots, \boldsymbol{x}_b)$. The conditional likelihood for any sample $\boldsymbol{x}_j$ is independent of $\boldsymbol{x}_{-j}$ conditioned on its neighbors (which are $\boldsymbol{x}_{-j}$), so

$$\mathcal{L}_{\text{train}}(\{x_j\}_{j=1}^b; \boldsymbol{\theta}) = \sum_{j=1}^b \log \tilde{p}_{\boldsymbol{\theta}}(\boldsymbol{x}_j|\boldsymbol{x}_{-j})$$

is the pseudo-log-likelihood objective (Besag, 1975) for $\{x_j\}_{j=1}^b$, which approximates the log-likelihood of the joint distribution. $\qquad\square$

## B  FOOLING LIKELIHOOD-BASED PERMUTATION TESTS

As suggested by empirical evidence, we assume that model mis-specification always occurs and that our likelihood estimates cannot be used naively for detecting out-of-distribution samples due to the mode seeking nature of $D_{\text{KL}}$. Under this assumption, likelihood-based permutation tests may seems more effective than simple thresholding rules for detecting out-of-distribution samples, as it treats both high-likelihood and low-likelihood samples as OoD. In this section, however, we show that likelihood-based permutation test can also be easily fooled, making them infeasible for OoD detection in a straightforward manner.

We consider fooling a model $p_{\boldsymbol{\theta}}(\boldsymbol{x})$ using samples generated from another model $q_{\boldsymbol{\phi}}(\boldsymbol{x})$ (trained on another dataset), where we modify the "temperature" $T$ of the generative model (Kingma & Dhariwal, 2018) (see Appendix C.2 for details). Then we gather the likelihood estimates for samples in $p(\boldsymbol{x})$ and $q(\boldsymbol{x})$ and obtain the area under the ROC curve (AUC) using $T_{\text{perm}}$ as prediction values to classify if $\boldsymbol{x}$ is OoD. We consider a RealNVP (Dinh et al., 2016) model trained on CIFAR and a PixelCNN++ (Salimans et al., 2017) model trained on Fashion MNIST. We use a RealNVP trained on SVHN, and a PixelCNN++ trained on MNIST to fool the two $p_{\boldsymbol{\theta}}(\boldsymbol{x})$ models respectively. For RealNVP we set $T = 1.10$ and for PixelCNN++ we set $T = 1.17$.

In Figure 4, we show that it is possible to produce a distribution $q_{\boldsymbol{\phi}}(\boldsymbol{x})$ with generative models such that its likelihood distribution evaluated by $p_{\boldsymbol{\theta}}(\boldsymbol{x})$ lies around the median likelihood in $p(\boldsymbol{x})$. This results in low AUC and thereby confusing the test based on $T_{\text{perm}}$. We show some out-of-distribution sample with similar bits-per-dimension (BPD) to training samples in Figure 5, where the generated samples look drastically different from training samples visually, even though they have similar log-likelihoods. We note that flipping the classifier predictions (i.e. using $-T_{\text{perm}}$) will not address this issue, since we would then treat low likelihood samples as more likely to be in-distribution.

## C  ADDITIONAL EXPERIMENTAL DETAILS

### C.1  MODEL ARCHITECTURES

For RealNVP (Dinh et al., 2016), we consider downscale once, with 4 residual blocks for each affine coupling layer and 32 channels for each convolutional layer, except for CIFAR where we have 8 residual blocks with 64 channels for each convolutional layer.

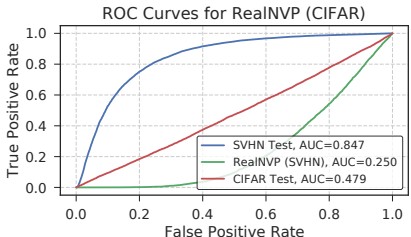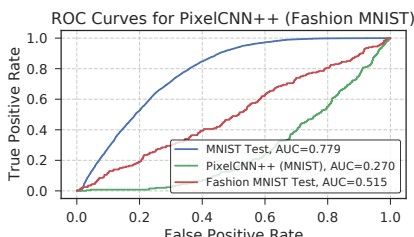

Figure 4: ROC curves for using the $p$-values for likelihood-based permutation tests. We assign positive labels to samples in $q(\boldsymbol{x})$ and negative labels to samples in $p(\boldsymbol{x})$. While such tests can detect OoD samples from other datasets, they could be confused by samples from another generative model.

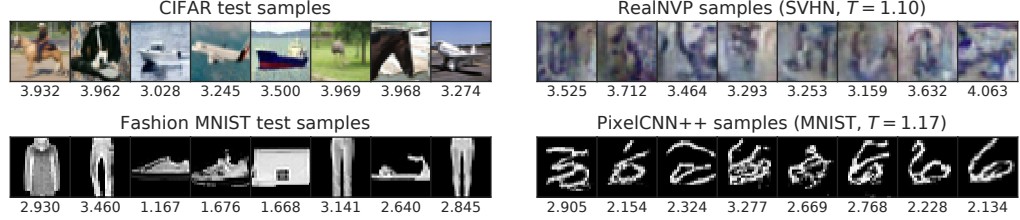

Figure 5: Samples and their BPD evaluated under $p_{\boldsymbol{\theta}}(\boldsymbol{x})$. (Top) $p_{\boldsymbol{\theta}}(\boldsymbol{x})$ is a RealNVP trained on CIFAR. (Bottom) $p_{\boldsymbol{\theta}}(\boldsymbol{x})$ is a PixelCNN++ trained on Fashion MNIST.

For VAE (Kingma & Welling, 2013; van den Berg et al., 2018), our inference network has 6 convolutional layers followed by a fully connected layer, and the generator network has 8 deconvolutional layers. We add batch normalization to all the convolutional layers and deconvolutional layers, except for the last two deconvolutional layers. We detail the convolutional layer hyperparameters in Table 3.

Table 3: Architecture for VAE convolutional layers. $k$ denotes image width divided by 4, $c$ denotes the number of channels of the image (1 for gray, 3 for colored).

| Inference network | | | | |
|---|---|---|---|---|
| Layer | Channels | Kernel | Stride | Padding |
| 1 | $32c$ | 5 | 1 | 2 |
| 2 | $32c$ | 5 | 2 | 2 |
| 3 | $32c$ | 5 | 1 | 2 |
| 4 | $64c$ | 5 | 2 | 2 |
| 5 | $64c$ | 5 | 1 | 2 |
| 6 | 256 | $k$ | 1 | 0 |
| Generator network | | | | |
| Layer | Channels | Kernel | Stride | Padding |
| 1 | $64c$ | 5 | 1 | 2 |
| 2 | $64c$ | 5 | 2 | 2 |
| 3 | $64c$ | 5 | 1 | 2 |
| 4 | $32c$ | 5 | 2 | 2 |
| 5 | $32c$ | 5 | 1 | 2 |
| 6 | $32c$ | 5 | 1 | 2 |
| 7 | 256 | 5 | 1 | 2 |
| 8 | $256c$ | 1 | 1 | 0 |

For PixelCNN++, we made two simple modifications to default hyperparameters (Salimans et al., 2017). We reduce the number of filters in each ResNet block from 160 to 80, and we use batch normalization after convolution as opposed to adding weight normalization over convolution.

All the models are trained with default optimizer hyperparameters with a batch size of 64. We use a batch size of 64 also in evaluation mode.

## C.2 EXPERIMENTAL PROCEDURES FOR SECTION B

We consider controlling the entropy of the generated samples by controlling a temperature hyperparameter $T$. In RealNVP, this is realized by multiplying the latent variable by a factor of $T$, which is equivalent to sampling from $\mathcal{N}(0, T^2 \boldsymbol{I})$. In PixelCNN++, this is realized by dividing each pre-softmax scalar output by $T$; larger $T$ values would lead to higher entropy samples.

## C.3 LIKELIHOOD DIFFERENCES BETWEEN TRAINING MODE AND TESTING MODE ON GENERATED DATA

We consider measuring the likelihood differences between *training mode* and *testing mode* using a RealNVP trained on CIFAR. The samples are generated from a RealNVP trained on SVHN, with several temperatures $T \in \{0.7, 1.0, 1.3\}$. Similar to the observation with SVHN samples, there is a large gap between *training mode* and *evaluation mode* evaluations.

Table 4: Log-likelihood (measured in bits per dimension) calculated with a RealNVP trained on CIFAR and evaluated on generated samples from a RealNVP trained on SVHN. We report likelihood results with *training mode*, *evaluation mode*, and their difference ($\Delta$).

| Dataset | Mode | BPD | $\Delta$ |
|---|---|---|---|
| CIFAR | *evaluation* | 3.48 | 0.03 |
| | *training* | 3.51 | |
| SVHN | *evaluation* | 2.44 | 8.56 |
| | *training* | 11.10 | |
| RealNVP (SVHN, $T = 1.0$) | *evaluation* | 2.88 | 325 |
| | *training* | 328 | |
| RealNVP (SVHN, $T = 0.7$) | *evaluation* | 1.51 | 315 |
| | *training* | 317 | |
| RealNVP (SVHN, $T = 1.3$) | *evaluation* | 5.43 | 10.57 |
| | *training* | 16.0 | |

## C.4 OUT-OF-DISTRIBUTION DETECTION WITH ALTERNATIVE VALUES OF $r_1$ AND $r_2$

We include additional results with alternative values of $r_1$ and $r_2$. If our method is able to achieve high performance with small $r_2$, this suggests that we can detect the OoD examples realiably even as they occupy a small portion within the batch. We consider $r_1 \to 0$ (which is *evaluation mode*) and $r_2 \in \{0.15, 0.3, 0.5, 0.9\}$; we show our results in Table 5.

## C.5 HISTOGRAMS OF LOG-LIKELIHOODS

We further show the histograms of log-likelihoods of several test datasets when the model is a RealNVP trained on CIFAR-10 datasets in Figure 6; the mean BPD is computed with evaluation mode.

## D DETAILS FOR THE DERIVATION IN SECTION 4.2.1

Suppose we try to learn some distribution $p(\boldsymbol{x})$ with a 2-d flow model with one coupling layer (Dinh et al., 2016): $f : (x_1, x_2) \mapsto (z_1, z_2)$ where

$$z_1 = x_1 \tag{12}$$

$$z_2 = x_2 + (x_1 - E(x_1))/\sqrt{V(x_1)} \cdot \gamma + \beta \tag{13}$$

Table 5: Out-of-distribution classification evaluated with AUC (left) and Average Precision (right). Rotation denotes $q(\boldsymbol{x})$ uses images in $p(\boldsymbol{x})$ yet randomly rotate each image by $d \in (90, 270)$ degrees.

| $p(\boldsymbol{x})$ | Model | $q(\boldsymbol{x})$ | $r_2 = 0.15$ | $r_2 = 0.3$ | $r_2 = 0.5$ | $r_2 = 0.9$ |
|---|---|---|---|---|---|---|
| Fashion MNIST | RealNVP | Rotation | 0.93 / 0.95 | 0.91 / 0.94 | 0.98 / **0.99** | **0.99 / 0.99** |
| | | MNIST | 0.92 / 0.93 | 1.00 / 1.00 | 1.00 / 1.00 | **1.00 / 1.00** |
| | | Omniglot | 0.96 / 0.97 | 0.95 / 0.96 | 1.00 / 1.00 | **1.00 / 1.00** |
| | | KMNIST | 0.86 / 0.88 | 1.00 / 1.00 | 1.00 / 1.00 | **1.00 / 1.00** |
| Fashion MNIST | VAE | Rotation | 0.88 / 0.91 | 0.94 / 0.95 | 0.94 / 0.95 | **0.97 / 0.98** |
| | | MNIST | 0.86 / 0.88 | 0.99 / 0.99 | 0.56 / 0.64 | **1.00 / 1.00** |
| | | Omniglot | 0.93 / 0.95 | 0.98 / 0.99 | 0.90 / 0.83 | **1.00 / 1.00** |
| | | KMNIST | 0.85 / 0.88 | 0.99 / 0.99 | 0.84 / 0.87 | **1.00 / 1.00** |
| CIFAR | RealNVP | Rotation | 1.00 / 1.00 | 1.00 / 1.00 | 1.00 / 1.00 | **1.00 / 1.00** |
| | | SVHN | 0.92 / 0.90 | 1.00 / 1.00 | 1.00 / 1.00 | **1.00 / 1.00** |
| | | ImageNet | 0.74 / 0.72 | 0.92 / 0.90 | 0.98 / 0.98 | **0.98 / 0.97** |
| | | LSUN | 0.77 / 0.75 | 0.98 / 0.97 | 1.00 / 1.00 | **0.99 / 0.98** |
| CIFAR | PixelCNN++ | Rotation | 0.87 / 0.83 | 0.97 / 0.94 | 1.00 / 0.99 | **0.99 / 0.99** |
| | | SVHN | 0.76 / 0.72 | 0.95 / 0.92 | 1.00 / 1.00 | **0.99 / 0.99** |
| | | ImageNet | 0.62 / 0.58 | 0.74 / 0.70 | 0.87 / 0.85 | **0.89 / 0.87** |
| | | LSUN | 0.72 / 0.69 | 0.80 / 0.75 | 0.98 / 0.98 | **0.98 / 0.97** |

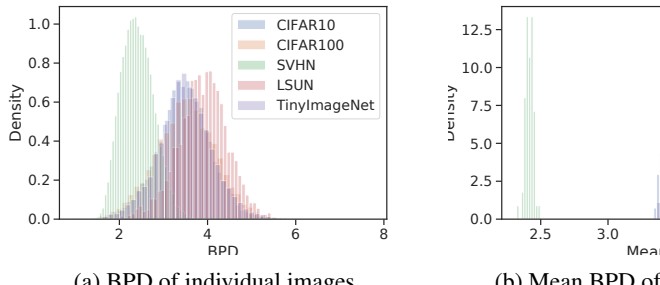

(a) BPD of individual images.   (b) Mean BPD of batch of images.

Figure 6: Histograms of BPD of test datasets on a RealNVP dataset trained on CIFAR-10.

with learnable parameters $\gamma, \beta$; $E(x_1), V(x_1)$ are the empirical mean and variance of a batch of $x_1$ in *training mode*, and the mean and variance of $p(x_1)$ in *evaluation mode*.

We assume the distribution $p(x_1)$ to be a even mixture of two Gaussians ($\mathcal{N}(-1, \sigma^2)$ and $\mathcal{N}(1, \sigma^2)$ with $\sigma \to 0$, so it concentrates around $-1$ and $1$), and $p(x_2) = \delta_0(x_2)$ ($x_2$ concentrates at zero), so $z_2 \approx (x_1 - E(x_1))/\sqrt{V(x_1)} \cdot \gamma + \beta$.

The Jacobian of $f$ is:

$$J = \frac{\partial f(\boldsymbol{x})}{\partial \boldsymbol{x}} = \begin{bmatrix} 1 & 0 \\ \gamma/\sqrt{V(x_1)} & 1 \end{bmatrix} \tag{14}$$

so its determinant is one. The likelihood of the flow model is simply

$$p_f(x_1, x_2) = p(f(x_1, x_2))|J| = p(f(x_1, x_2))$$

We assume the prior $p(z_1, z_2)$ is standard Gaussian $\mathcal{N}(0, I)$. Therefore, we have:

$$p_f(x_1, x_2) \approx \varphi(x_1) \cdot \varphi((x_1 - E(x_1))/\sqrt{V(x_1)} \cdot \gamma + \beta) \tag{15}$$

where $\varphi(x)$ is the probability density function for the standard Gaussian distribution. Note that $\gamma, \beta$ only depends on the second term. The maximum likelihood solution for $\beta = 0$ (due to symmetry); for $\gamma$, we can treat $p(x_1)$ as two point distributions at $-1$ and $1$, so we have the optimal $\gamma \approx \arg\min_s -s^2/2 + \log s = 1$. Moreover, $\mathbb{E}_p[x_1] = 0$ and $\text{Var}_p[x_1] \approx 1$.

For $p(x_1, x_2)$ in evaluation mode, we have

$$\mathbb{E}_p[\log p_f(x_1, x_2)] \approx \mathbb{E}_p[\log \varphi(x_1) + \log \varphi((x_1 - E(x_1))/\sqrt{V(x_1)}] \tag{16}$$
$$= \mathbb{E}_p[\log \varphi(x_1) + \log \varphi(x_1)] \approx -2.84. \tag{17}$$

For $q(x_1, x_2)$ in evaluation mode, we have

$$\mathbb{E}_q[\log p_f(x_1, x_2)] \approx \mathbb{E}_q[\log \varphi(x_1) + \log \varphi(x_1)] \approx -1.92. \tag{18}$$

For $q(x_1, x_2)$ in training mode, $E(x_1) = 0$, $V(x_1) = 1/\sqrt{12}$, so we have:

$$\mathbb{E}_q[\log p_f(x_1, x_2)] = \mathbb{E}_q[\log \varphi(x_1) + \log \varphi(\sqrt{12}x_1)] \approx -2.38. \tag{19}$$

