# OpenReview forum: "Unsupervised Out-of-Distribution Detection with Batch Normalization"
_ICLR.cc/2020/Conference — Reject_

### Official Review · AnonReviewer2 · 2019-10-22
**Official Blind Review #3**

**Rating:** 1

**Review:**

This paper attempts to address the problem of out-of-distribution detection with generative models. To do this they assume they are given batches of OOD examples or batches of in-distribution examples, and they detect whether the batch is in- or out-of-distribution. Normally we try to detect if an example is in- or out-of-distribution.
Unfortunately, this is not an interesting assumption and makes the problem significantly easier. If we assume this, then averaging the anomaly scores of multi-class OOD detectors would result in performance near the ceiling.
It is not surprising that one can obtain a much better OOD detector given a batch of OOD samples. Hence they're making progress on a problem we have the community has not been interested in, and they are not making progress on the standard OOD detection problem.

Small notes:

> given their successful generalizing on a test dataset.
What does this mean? Does this mean their BPP is good? By what standard?

> unerlying
underlying

> CIFAR
Include CIFAR-10 vs CIFAR-100 results in the table.


**Experience Assessment:**

I have published in this field for several years.

**Review Assessment: Checking Correctness Of Derivations And Theory:**

I assessed the sensibility of the derivations and theory.

**Review Assessment: Checking Correctness Of Experiments:**

I carefully checked the experiments.

**Review Assessment: Thoroughness In Paper Reading:**

I read the paper at least twice and used my best judgement in assessing the paper.

---

### Official Review · AnonReviewer1 · 2019-10-24
**Official Blind Review #1**

**Rating:** 6

**Review:**

This paper tackles the out of distribution detection problem and utilizes the property that the calculation of batch-normalization is different between training and testing for detecting out-of-distribution data with generative models. The paper first empirically demonstrates that the likelihood of out-of-distribution data has a larger difference between training mode and testing mode, then provides a possible theoretical explanation for the phenomenon. The proposed scoring function utilizes such likelihood differences in a permutation test for detecting out-of-distribution data. The evaluation is performed on two small in-distribution image datasets and four out-of-distribution datasets with three types of generative models.

We recommend a weak accept. Its clarity is good, and the strength of the paper has three parts. The first is the thorough observation of the likelihood changes between different modes of batch-normalization. Second, the theoretical explanation for the observed phenomenon is sound. The example in Figure 2 gives a good intuition of how mis-specification can happen. The last is the strong performance on the out-of-distribution detection with generative models.

However, I have some concerns about the design of the scoring function (Section 5), which looks like it is carefully tuned:
1. Why do the authors use the permutation score ($T_{b,r1,r2}$) instead of likelihood difference ($\delta_{b,r1,r2}$)? The likelihood difference itself seems like it could be a good indication for OoD detection.
2. Why do the authors use interpolation between training and evaluation? It introduces extra hyperparameters (r1, r2) for the method. Is the performance sensitive to the choice of r1 and r2?

One minor issue:
3. The sentence at the bottom of page 7 should be removed ("related work still needs some work, but there seems to be some bug on overleaf right now")


**Experience Assessment:**

I have read many papers in this area.

**Review Assessment: Checking Correctness Of Derivations And Theory:**

I assessed the sensibility of the derivations and theory.

**Review Assessment: Checking Correctness Of Experiments:**

I assessed the sensibility of the experiments.

**Review Assessment: Thoroughness In Paper Reading:**

I read the paper thoroughly.

---

### Official Review · AnonReviewer3 · 2019-10-25
**Official Blind Review #3**

**Rating:** 1

**Review:**

The paper makes the observation that likelihood models trained with batch norm assign much lower likelihoods to "training batches" of OoD data (batch norm statistics computed over over minibatch) than evaluation batches of OoD data (batch norm statistics over entire training set).

One issue with comparing this method to most other OoD detection works is that it considers OoD detection on *batches* of (all OoD data) or (all in-distribution data). As soon as the problem is changed to "classify between OoD batches" and not single samples, there are a large number of possible statistical tests one can perform to perform OoD (T-test between likelihoods of each batch) and the problem becomes *much* easier. In some ways, this makes things more well-defined (hard to compare distributions when one of them is just a single sample from an arbitrary distribution).

However, that brings me to a big concern I have with the evaluation protocol. The batch size used for train/evaluation is rather large (64, this detail is hidden in the Appendix and I would have appreciated the number put in the main experiments section). If you take a likelihood model and evaluate on 64 samples from SVHN, you are all but guaranteed to sample a sample with *exceedingly* low likelihood, which dominates the mean statistic, making it possible to separate SVHN batch from CIFAR10 batches. I suspect that OoD datasets have plenty of these "extremely low likelihood" examples that will drag the mean likelihood down a lot.

This is consistent with your batch normalization experiments: in training mode, the likelihood is computed from mean activations over a batch of OoD samples, several of which probably contribute to the low likelihoods. In evaluation mode, likelihoods for each OoD sample are evaluated independently, which results in a similar observation to prior work showing that CIFAR10 likelihoods are inaccurate for SVHN. In other words, I think there is a mistake made here: it is the phenomenon that *batch likelihoods*, not *batch norm*, that is responsible for this method working well. One experiment that is missing from your paper (and would prove my hypothesis wrong) would be if you adapted the OoD criteria to compare the *mean* likelihoods in the evaluation mode, and show that for OoD datasets, the difference between batches still remains small.

Nits:
- "...such as learning a mixture of Gaussians", I believe this toy example was on univariate gaussians, not mixtures.
- Choi et al. 2018 and should also be included in the citation that "CIFAR10 gives higher likelihood estimates to SVHN than CIFAR10 ones" (this was a concurrent discovery between the two papers)
- Choi et al. 2018 is not the right citation for "we evaluate the area under the ROC curve (AUC) and average precision (AP)" for each binary classification task, a more appropriate one would be Hendryks and Gimpel 2017.
- No doubt the authors realized already but Page 7 has some "related work still needs some work, but there...." which should be deleted.
- It took me awhile to find the batch size used in training and evaluation mode (64), which was on page 16.

**Experience Assessment:**

I have published one or two papers in this area.

**Review Assessment: Checking Correctness Of Derivations And Theory:**

I assessed the sensibility of the derivations and theory.

**Review Assessment: Checking Correctness Of Experiments:**

I carefully checked the experiments.

**Review Assessment: Thoroughness In Paper Reading:**

I read the paper thoroughly.

---

### Decision · Program_Chairs · 2019-12-19

**Decision:**

Reject

**Comment:**

The authors observe that batch normalization using the statistics computed from a *test* batch significantly improves out-of-distribution detection with generative models.  Essentially, normalizing an OOD test batch using the test batch statistics decreases the likelihood of that batch and thus improves detection of OOD examples.  The reviewers seemed concerned with this setting and they felt that it gives a significant advantage over existing methods since they typically deal with single test example.  The reviewers thus wanted empirical comparisons to methods designed for this setting, i.e. traditional statistical tests for comparing distributions.  Despite some positive discussion, this paper unfortunately falls below the bar for acceptance.  The authors added significant experiments and hopefully adding these and additional analysis providing some insight into how the batchnorm is helping would make for a stronger submission to a future conference.